# Is It Time to Change the Definition of Acute Exacerbation of Chronic Obstructive Pulmonary Disease? What Do We Need to Add?

**DOI:** 10.3390/medsci6020050

**Published:** 2018-06-14

**Authors:** Maria Montes de Oca, Maria Eugenia Laucho-Contreras

**Affiliations:** 1Hospital Universitario de Caracas, Facultad de Medicina, Los Chaguaramos, Universidad Central de Venezuela, 1053 Caracas, Venezuela; 2Servicio de Neumonología, Hospital Universitario de Caracas, Facultad de Medicina, Universidad Central de Venezuela, 1053 Caracas, Venezuela; mlaucho@gmail.com

**Keywords:** Chronic obstructive pulmonary disease, acute exacerbation of chronic obstructive pulmonary disease, biomarkers, symptom-related definition, healthcare-based definition

## Abstract

Acute exacerbations in chronic obstructive pulmonary disease (AECOPD) are associated with increased mortality, rate of hospitalization, use of healthcare resources, and have a negative impact on disease progression, quality of life and lung function of patients with chronic obstructive pulmonary disease (COPD). There is an imperative need to homogenize the definition of AECOPD because the incidence of exacerbations has a significant influence or implication on treatment decision making, particularly in pharmacotherapy and could impact the outcome or change the statistical significance of a therapeutic intervention in clinical trials. In this review, using PubMed searches, we have analyzed the weaknesses and strengths of the different used AECOPD definitions (symptom-based, healthcare-based definition or the combinations of both), as well as the findings of the studies that have assessed the relationship of different biomarkers with the diagnosis, etiology and differential diagnosis of AECOPD and the progress towards the development of a more precise definition of COPD exacerbation. Finally, we have proposed a simple definition of AECOPD, which must be validated in future clinical trials to define its accuracy and usefulness in daily practice.

## 1. Introduction

Chronic obstructive pulmonary disease (COPD) is a highly prevalent disease; its natural history is often associated with episodes of acute worsening of symptoms, and it is accompanied by a variable degree of physiological deterioration known as exacerbations. They are associated with increased mortality, rate of hospitalization, use of healthcare resources, and have a negative impact on disease progression, quality of life and lung function of patients with COPD [1,2,3,4]. For these reasons, documents such as the Global Initiative on Chronic Obstructive Lung Disease (GOLD) have stated that the prevention and treatment of exacerbations should be a key goal of COPD management [5]. As a consequence, the incidence of exacerbations has significant influence or implication on treatment decision-making, in particular on pharmacotherapy.

Data from the Evaluation of COPD Longitudinally to Identify Predictive Surrogate Endpoints (ECLIPSE) study identified a subgroup of individuals, known as “frequent exacerbators”, defined as those who had two or more exacerbations annually, which represents 12% of the study population [6]. Since then, “frequent exacerbators” has been accepted as a COPD phenotype and incorporated into the current GOLD multidimensional assessment of COPD [5]. Additionally, when patients with COPD GOLD-1 are included in the population studied, the prevalence of this phenotype decreases to 2%. The most common phenotype was no-exacerbators (51%) followed by inconsistent exacerbations (both years with and without events, 41%) [7].

Recently, several studies in the COPD population have reported the distribution of patients according to the new GOLD-2017 classification (A, B, C and D groups). Data from the population-based Latin-American Pulmonary Obstruction Investigation Project (PLATINO) study indicates that patients were distributed as follows: 69% in group A, 26% in group B, 2% in group C, and 3% in group D [8]. Similar distribution was found in another population-based study from Argentina (A: 52%, B: 43%, C: 1% and D: 4%) [9].

In a selected COPD cohort from Spain and the United States, Cabrera et al. reported that only around 20% of the population were at high risk of exacerbation (8.2% group C and 11.7% group D) [10]*.* Similar results have been reported in another selected COPD population in which approximately half of the patients classified as D (GOLD-2011) changed to B (GOLD-2017). Patients categorized as GOLD-B (2017) exacerbated 17% more than GOLD-B (2011) [11].

The results of all the studies suggest that the new GOLD classification significantly increases the proportion of patients in the groups A and B or at low risk of exacerbation, whereas it decreases the proportion of patients categorized in groups C and D or at high risk of exacerbation.

Defining acute exacerbations in chronic obstructive pulmonary disease (AECOPD) is important not only in daily clinical practice, but also for research. However, there is a lack of consensus regarding the definition of AECOPD and an absence of objective measurements in the definitions currently used. Many definitions have been based on the presence of symptoms (symptom-based definition), the types of healthcare resources used (healthcare-based definition), or the combinations of both.

Currently, in most of the international guidelines the diagnosis of an exacerbation relies exclusively on the clinical presentation of the patient complaining of acute changes of symptoms, often confounded by symptoms associated with comorbidities; there is no biomarker or panel of biomarkers that help to make a more accurate diagnosis.

In this review, we discuss the different proposed AECOPD definitions, their impact on effect sizes and therapeutic decision-making, the role of physiological and other biomarkers in the diagnosis of AECOPD, and the progress towards the development of a more precise definition of COPD exacerbation.

## 2. Symptom-Based Definition

One of the most referenced definitions of AECOPD is the classic Anthonisen et al. [12] definition, which represent the basis of many subsequent criteria. The definition was developed for investigating the use of antibiotics in COPD exacerbations based on the presence of one or more of three cardinal symptoms (increase or new onset of breathlessness, sputum production and sputum purulence). This described three levels of exacerbation (Table 1). The type 1 when the three symptoms (increased dyspnea, sputum volume and sputum purulence) were present; the type 2 when two of these symptoms were present, and type 3 when one of the three symptoms was present in addition to at least one of the following: upper respiratory infection within the past five days, fever without other cause, increased wheezing or cough, and an increase in respiratory rate or heart rate by 20% compared with baseline. The results of the study indicate that antibiotics should be indicated in those exacerbations presenting with the three cardinal symptoms, whereas that type-3 do not benefit with the use of these drugs [12]. Antibiotic therapy is probably justified when two of the three major symptoms are present (success rate with antibiotics 70% vs. 60% with placebo) [12].

Another symptom-based definition was described by Rodriguez-Roisin [13] that defined an AECOPD as “a sustained worsening of the patient’s condition from the stable state and beyond normal day to day variations, that is acute in onset and necessitates change in regular medication in a patient with underlying COPD” (Table 1). This definition was partially adopted by a previous GOLD version in which an AECOPD was defined as “an acute event characterized by a worsening of the patient’s respiratory symptoms that is beyond normal day-to-day variations and leads to a change in medication” [23]. The new GOLD-2017 document had simplified the definition as “an acute worsening of respiratory symptoms that results in additional therapy” (Table 1) [5].

The Spanish guidelines for diagnostic and treatment of COPD (GesEPOC) define AECOPD as “a clinical episode occurring during the course of COPD, characterized by a sudden or gradual worsening of symptoms that is beyond expected daily variability and cannot be attributed to other disorders” [14]*.* On the other hand, the recent European Respiratory Society/American Thoracic Society (ERS/ATS) guideline on the management of COPD exacerbations has proposed another symptom-based definition of exacerbations: “episodes of increasing respiratory symptoms, particularly dyspnea, cough and sputum production, and increased sputum purulence” [15].

Although the above-proposed definitions are simple and may be useful in clinical practice, none of them include objective measurements nor have they been validated in subsequent studies. As the patient’s symptomatology varies widely, and a precise level of worsening of dyspnea or increased sputum production has not been defined as a diagnostic of exacerbation, an objective assessment of “symptom’s worsening” and a validation of the magnitude of the changes in symptom scales are still required. In addition, these definitions do not allow differentiating worsening of symptoms due to exacerbation from other common comorbidities in COPD patients, such as acute coronary syndrome, worsening congestive heart failure, pulmonary embolism and pneumonia. These conditions must be excluded as causes for clinical worsening of COPD, and the definition of AECOPD should include measurements that make it possible. Therefore, it has been necessary to move forward the analysis and the development of more objective tools or measurements that help to define relevant changes in symptoms related with COPD exacerbation and improve the accuracy of the symptom-based definition.

Different studies have evaluated the usefulness of the patient-reported outcomes (PROs) to improve the accuracy of the definition of exacerbations based on the symptoms and the detection of the relevant changes in symptoms related with AECOPD.

The Food and Drug Administration (FDA) in its guidance defines a PRO as “any report of the status of a patient’s health condition that comes directly from the patient, without interpretation of the patient’s response by a clinician or anyone else” and has highlighted the importance and value of PRO measures for assessing the treatment efficacy in COPD clinical trials [24]*.*

The most extensively validated PROs in exacerbation studies are the exacerbations of chronic obstructive pulmonary disease (EXACT) [25,26]. This daily symptom diary seems to be useful for assessing exacerbation frequency, duration, and severity, and has been qualified as an exploratory endpoint by the FDA and the European Medicines Agency (EMA). Validation studies document that this PRO reliably assesses symptom severity and that EXACT scores are significantly elevated at exacerbation compared to stable state values (53, 55) [25,27].

Although PROs seem to be useful tools to improve the accuracy of the AECOPD definitions, they have limitations in clinical practice that surround mainly copyright ownership for instruments such as the EXACT that would preclude mass usage due to cost, and fatigue from patients that might reduce completion rates if they are used over long periods of time.

## 3. Healthcare-Based Definition

Although symptom-based definitions are commonly adopted, healthcare or event-based definitions, including change of therapy (antibiotics and/or oral corticosteroids) or management (emergency room attendance or hospital admission), have been also used, in particular in clinical trials to solve the problems of the symptom-based definition [16,17,18,19,20,21,22]. Therefore, the main objective of these definitions is to identify the patients whose condition has changed enough to require changes of medical treatment (requirement for oral steroids or antibiotics) or hospitalization. Some examples of event-based or a combination of symptom and healthcare-based definitions used in clinical trials are presented in Table 1.

Some studies have reported that only around 50% of all AECOPD identified by worsening symptoms are reported to healthcare professionals for treatment, so healthcare or event-based definitions appear to identify fewer events than symptom-based definitions and are likely to select a distinct group of patients [28,29].

Although, event-based definitions appear to be more objective than symptom-based definitions and provide a more direct approach, they continue to have flaws, as well as a lack of standardization and validation for their general use in clinical practice.

One of the main difficulties in the event-based definitions of exacerbations is related to the wide variations of the different schemes recommended for treating the AECOPD and the allocation or availability of resources among the different health systems. Even within the same health systems, there is great variability among doctors in the decision-making and therapeutic prescription pattern for AECOPD. Another problem is the self-prescription of antibiotics and/or systemic corticosteroids by the patient (particularly in countries where these drugs are available over the counter), which could be a common practice in frequent exacerbators. Therefore, the event-based definitions of exacerbations are complicated and are limited by the considerable variability and heterogeneity of factors associated to the patients, physicians’ behavior, healthcare systems and the inability to detect unreported events. One way to improve the accuracy of the AECOPD definition based on events would be the incorporation of information collected directly from patients with standardized instruments such as PROs in combination with the medical verification of events [30].

## 4. Impact of Acute Exacerbations in Chronic Obstructive Pulmonary Disease: Definition on Effect Size and Therapeutic Decision-Making

### 4.1. Impact on the Effect Size of Therapeutic Interventions in Clinical Trials

Currently, the frequency of AECOPD has become an important primary outcome to be measured in the context of COPD clinical trials due to their great burden for the patient, and the high economic costs to society.

It has been reported in clinical trials that the decision on which definition of AECOPD to apply can have an impact on the outcome and can affect the number of exacerbations observed [31]. Effing et al. assessed the potential impact of AECOPD definition on the size and significance of treatment effect (relative risk and hazard ratio) in a randomized controlled discontinuation trial of inhaled corticosteroids (ICS) [31]. Six definitions of AECOPD and two effect parameters (number of exacerbations per year and time to first exacerbation) were used for the analyses. Applying the different definitions of AECOPD, the relative risks (RRs) for the exacerbation rate ranged from 1.19 to 1.49, and hazard ratios (HRs) for time to first exacerbation ranged from 1.36 to 1.84 for the various definitions, varying from non-significant to significant [31]. The highest (statistically significant) RRs and HRs were found with the definitions based on treatment with oral corticosteroids and/or antibiotics and exacerbation according to Anthonisen et al. [31]. The authors conclude that different definitions of an AECOPD led to different effect sizes and hence different conclusions about the effect of withdrawing ICS in patients with COPD.

The result of this study indicates that a change in the definition may affect the number of exacerbations and could change the statistical significance of a therapeutic intervention. Therefore, in the analysis of the clinical trials results is imperative to have a clear description of the definition used for AECOPD and to take it into account when making comparisons with other studies. The lack of a precise and standardized definition of AECOPD makes complex the comparison of the treatment effect among different studies. The inclusion of standardized validated measures such as the PROs combined with investigator assessment to evaluate COPD exacerbations in prospective clinical trials could facilitate the comparison of the results among the studies [30]

### 4.2. Impact of the Definition of Exacerbations in the Therapeutic Decision-Making

In the clinical setting, defining AECOPD is important because it has implications on the decision-making of the type of medical treatment recommended for a particular patient.

The vast majority of current COPD guidelines recommend guiding medical treatment mainly towards to the reduction of symptoms and the future risk of exacerbations. Therefore, it is important to discuss the potential pharmacotherapy implications associated with the determination of the future risk of exacerbations.

The new GOLD-2017 document indicates that the best predictor of frequent exacerbations (defined as ≥2 exacerbations per year) is a history of earlier treated events [5]. This document states that patients should receive preventative therapy for exacerbations if they have experienced ≥2 exacerbation or ≥1 exacerbation leading to hospital admission within the previous 12 months [5]. In patients categorized as group C (less symptoms, high risk of exacerbation), it recommends initiating pharmacological therapy with a long-acting muscarinic antagonist (LAMA) due to its superiority over long-acting β_2_ agonist (LABA) regarding exacerbation prevention, and to add a second long-acting bronchodilator (LABA/LAMA), or the use of LABA/ICS if the patient persists exacerbating [5]. In group D (more symptoms, high risk of exacerbation) it recommends initiating with a LABA/LAMA combination and escalating to triple therapy (LABA/LAMA/ICS) if the patient experiences further exacerbations [5]. 

As was discussed above, the diagnosis of an AECOPD according to GOLD document relies mainly on the clinical presentation of the patient complaining of an acute change of symptoms, often confounded by symptoms associated with comorbidities. Therefore, this may potentially result in inappropriate treatment with the combination of unnecessary medications (overtreatment) in some patients who are misclassified as high-risk (groups C and D), in particular those with mild to moderate airflow limitation (GOLD 1–2).

## 5. The Role of Biomarkers in the Diagnosis and Definition of AECOPD

A biomarker has been defined by the Biomarkers Definition Working Group as a characteristic that is objectively measured and evaluated as an indicator of normal biologic processes, pathogenic processes, or pharmacologic responses to a therapeutic intervention [32]. Theoretically, the clinical parameters could be used as biomarkers but in practice, the biomarkers are used as measurable, reproducible and objective tests that provide information about the state of a disease [33]. A biomarker could also help to identify “early” disease while it is still relatively asymptomatic, and to clarify the differential diagnosis. The definition of AECOPD by biomarkers is a challenging field, mainly due to the fact that studies have been conducted in heterogeneous populations, have poor reproducibility and limited external validation [34,35].

There are biomarkers at different cellular and subcellular levels in AECOPD (cellular, genetic, molecular, serum metabolomics or sputum transcriptomic proteins). Biomarkers provide information before, during and after the exacerbation. This might guide the treatment of COPD patients; decrease the use of unnecessary drugs and avoid their adverse effects. There is a wide range of samples studied, which can be categorized as local (lung samples) or systemic (blood samples). A recent systematic review [34] found that the main biomarkers measured are acute phase reactants (C-reactive protein (CRP), erythrocyte segmentation rate (ESR) and fibrinogen), cytokines (interleukin (IL)-6, IL-8 and TNF-α), molecules of cardiac origin (brain natriuretic peptide (BNP)), molecules involved in collagen formation (matrix metalloproteinase (MMP)-9) and molecules involved in fatty acid processing (adiponectin). The biomarkers can also be classified by their clinical use in COPD. The Table 2 summarizes the clinical use of biomarkers for the AECOPD.

### 5.1. Biomarkers Associated to the Definition and Risk of Exacerbation

CRP is the most frequently studied acute phase reactants (APR) in AECOPD. Several studies have reported a marked increase in CRP plasma levels during AECOPD [36,37,38,68,69,70]. However, there is a wide range in the CRP threshold values reported among the studies. Hurst et al. [36], found that plasma levels of CRP were associated with the diagnosis of exacerbation (cut-off 5 mg/L (74,4% sensitivity and 57,5% specificity for the diagnosis of AECOPD)), in the presence of clinical symptoms of exacerbation (increased dyspnea, sputum purulence or sputum volume). In this study, CRP level was not related with the exacerbation severity, probably because most of the patients studied had mild-moderate AECOPD. In the same line, other studies have confirmed that the addition of CRP levels to the exacerbation symptoms increased the predictive power for AECOPD (Reciever Operating Characteristics (ROC) > 0.8) [36,69,70]. In contrast, Karadeniz et al., in patients with more severe AECOPD [38] who had been admitted to a regular ward or in intensive care unit, found that increased CRP plasma level was related with the severity of the exacerbation [6.28 ± 6.53 mg/dL in the regular ward cases and 16.9 ± 12.03 mg/dL in the Intensive Care Unit (ICU) patients (*p* < 0.01)]. Moreover, Chen et al. [39] assessed the discriminatory power of blood CRP and N-Terminal-pro brain natriuretic peptide (NT-proBNP) in the diagnosis of AECOPD requiring hospitalizations. The authors concluded that a combinatorial approach could separate patients who were experiencing AECOPD that required hospitalization from stable patients. The authors replicated these findings in an external cohort. In the same line, other authors established the use of CRP as a prognosis marker by showing that a higher level of CRP is associated with in-hospital treatment failure in AECOPD [71].

In the ECLIPSE cohort, Hurst et al. [6], found that several plasma biomarkers [fibrinogen (mg/dL) OR 1.35, 95% confidence interval (CI) (1.22–1.49), high sensitivity (hs) CRP (mg/L) OR 1.24 95% CI (1.13–1.37), chemokine (C-C motif) ligand-18 (CCL-18) (ng/mL) OR 1.13, 95% CI (1.02–1.25) and surfactant protein-D (SP-D) (ng/mL) OR 1.10 95% CI (1.01–1.20)] and cellular biomarkers [platelet count (10 × 10^9^/L increase) OR 1.02 95% CI (1.01–1.04), white blood cell (WBC) count (1 × 10^9^/L increase) OR 1.07 95% CI (1.03–1.12) and neutrophil count (1 × 10^9^/L increase) OR 1.02 95% CI (1.01–1.03)] were significantly correlated with the risk of exacerbation in one year. However, after multivariable adjustment including previous exacerbations, only the elevated WBC remained statistically significant as a predictive biomarker of future exacerbations [6]. Supporting the role of WBC, a more recent study [40] found that the combination of plasma levels of high sensitive (hs) CRP (3 mg/L), fibrinogen (14 μMol/L) and WBC count (9 × 10^9^/L) were associated with increased risk of exacerbations in COPD patients, regardless of the disease severity and the exacerbation history.

Fibrinogen is a biomarker recurrently studied in COPD patients. Several trials showed that plasma fibrinogen levels are increased during exacerbations [41,42] (defined by the Anthonisen criteria [12]) in comparison with the stable state. However, fibrinogen has a better use as a potential biomarker for exacerbation risk [43,44] and mortality [45] in COPD patients rather than for defining of AECOPD.

Others biomarkers with elevated levels during exacerbations IL-6 [42,46], serum uric acid [48], surfactant protein-D (SP-D) [49], CCL-18 [50], vasoactive intestinal peptide [51], copectine [52] and e-selectine [53]. In addition, MMP-9/tissue inhibitor of metalloproteinase protein-1 (TIMP-1) is a protease/anti-protease pathway that has been studied in AECOPD. The findings of a study showed that levels of MMP-9 and TIMP-1 in exhaled breath condensate (EBC) [56] and in plasma [54] are increased during AECOPD and are negatively correlated with spirometric variables [56]. Furthermore, there is evidence in COPD patients with α one antitrypsin deficiency (A1ATD) that elevated plasma levels of MMP-9 are associated with worse lung function and increased risk of exacerbation compared to patients with lower levels [55].

### 5.2. Biomarkers Associated with the Etiology of Exacerbation

Several studies have documented the heterogeneity of the AECOPD, as well as its relationship with several biomarkers [72,73]. A prospective study identified four exacerbations clusters: bacterial, viral, eosinophilic, and the last cluster was related to minimal changes in the inflammatory response named “pauci-inflammatory” [73]. In addition, there is evidence indicating that the use of clinical parameters such as lung function, respiratory rate and dyspnea provide an objective description of the severity of the episode; meanwhile, the measurement of the inflammatory cytokines provides a better understanding of the underlying pathophysiology of the exacerbation [74].

Another molecule that indicated a bacterial etiology in AECOPD is the pentraxin-3 (PTX-3) [57]. PTX-3 is a soluble pattern recognition receptor, recognizing pathogen-associated molecular patterns expressed by microorganisms. In COPD patients, a study showed that levels PTX-3 in sputum modestly rise during the exacerbation, and there was a correlation with bacterial isolation in sputum (ROC area under the curve (AUC) 0.65, 95% CI 0.52–0.78, *p* = 0.03).

Other authors [47] reported that the addition of IL-6 to a symptoms-based definition of exacerbation could accurately identify the presence of AECOPD caused by viral infection (specificity 87–96% with 78% of true viral infections correctly identified). Plasma levels of interferon γ-induced protein-10 (IP-10) [58], soluble IL-5 receptor α [59] and monocyte chemotactic protein-1 (MCP-1) [46] have also been shown to be useful for discriminating viral infections in AECOPD.

Different studies have focused on the use of cytokines to identify bacterial infections in AECOPD. The findings of a study that analyzed the sputum cytokines levels in stable COPD, and in AECOPD determined that TNF α was the cytokine with best diagnostic accuracy to establish the bacterial etiology of AECOPD [60]. In contrast, plasma levels of TNF α have been liked with viral etiology of exacerbations [46]. 

On the other hand, a two-fold increase of IL-1β in the sputum with respect to stable COPD state has been reported in patients with AECOPD [61], and this was associated with a bacterial etiology. Furthermore, in the same study, the authors stimulated cultured cells from airway epithelium, smooth muscle, and lung endothelia with IL-1β and successfully induced TNF-α, granulocyte-colony-stimulating factor (G-CSF), IL-6, CD-40L, and MIP-1 in all three-airway structural cell types.

Procalcitonin (PCT) is a peptide associated with bacterial infection and sepsis. A study randomized COPD patients that required hospitalization for exacerbations in three groups: low levels [<0.1 μg/L (considered with absence of bacterial infection and the use of antibiotic was discouraged)], medium levels [between 0.1–0.25 μg/L (possible bacterial infection, and the use of antibiotic was discouraged or encouraged according to clinical criteria)] and high levels [>0.25 μg/L, (bacterial infection, and the use of antibiotics was encouraged)] [75]. In this study, PCT levels could reduce the use of antibiotics in hospitalized AECOPD patients without adverse outcome [75]. In the same line, other studies consistently showed the association between bacterial AECOPD and PCT levels [76,77]. PCT to direct the use of antibiotic therapy has been studied as part of the treatment algorithm of AECOPD in the emergency room [78]. A recent meta-analysis [62] showed that guidance with PCT levels (cut-off 0.25 μg/L) significantly limits the antibiotic exposure, without alteration of clinical outcomes (treatment success, re-exacerbation, re-hospitalization or mortality). However, the quality of the available data is low to moderate, suggesting the need for confirmatory clinical trials with rigorous methodology.

Eosinophils are a promising biomarker for the definition and classification of AECOPD. One third of all the exacerbations are controlled by a type-2 immunological response leading from eosinophils [73]. There is evidence of an increase in the eosinophil levels in sputum [60], cytokines related to eosinophilic and serum IL-5, chemokine (C-C motif) ligand 17 (CCL-17) and chemokine (C-C motif) ligand (CCL-13) [73] during exacerbation. Also, the levels of peripheral eosinophils have been linked with a lower risk of bacterial presence during an AECOPD [63].

Recently, the results of a population based study [64] showed that an eosinophil peripheral blood level above 0.34 × 10^9^ cells per liter in COPD patients increases the risk of exacerbations with multivariable-adjusted incident rate ratios (IRRs) of 1.76 (95% CI 1.56–1.99) for severe exacerbations and 1.15 (1.05–1.27) for moderate exacerbations. 

Additionally, a study investigates the usefulness of blood eosinophils to direct corticosteroid therapy during AECOPD. Blood eosinophils were measured in the biomarker-directed and standard therapy arms to define biomarker-positive and -negative exacerbations (blood eosinophil count > and ≤ 2%, respectively). The authors found that the use of corticosteroids guided by blood eosinophil level is a viable strategy with a little rate of treatment failure (<2%) in comparison with the non-directed approach that had a higher rate of failure (15%) [65]. Therefore, there is evidence indicating that the use of peripheral blood eosinophil count could be a biomarker to direct corticosteroid therapy during COPD exacerbations. Furthermore, there is evidence that the use of the absolute value of blood eosinophils instead of 2% of peripheral eosinophilia is more accurate and adequate for a diagnostic and therapeutic approach [79]. 

Currently, the controversies persist regarding the cut-off threshold for peripheral blood eosinophil count and the use of total count compared with the percentage of eosinophils for predicting the exacerbation risk as well as the ICS effects. Therefore, larger prospective clinical trials are required to clarify the blood eosinophil cut-off values that could be used in clinical practice.

### 5.3. Biomarkers Associated with Differential Diagnosis

One of the principal differential diagnoses of AECOPD is cardiac dysfunction. Therefore, biomarkers that help to identify cardiac disease during an exacerbation are needed. N-Terminal-pro-brain natriuretic peptide (NT-pro-BNP) is a polypeptide secreted by the cardiac ventricles in response to several stimulus [80]. There is evidence that increased levels of NT-pro-BNP (>220 pmol/L) and troponin T (>0.03 μg/L) strongly predict short- [81] and long-term [82] mortality in hospitalized patients with AECOPD. The International Collaborative Of NTproBNP (ICON) study [83] suggested using in all patients a general age-independent cut-off point of 300 pg/mL to exclude acute heart failure, whereas for diagnosis of heart failure, age-dependent cut-off points are probably more useful. In patients with COPD, an increase in cardiac biomarkers from the stable state to exacerbation were significantly higher in those patients with known ischemic heart disease compared with those without this condition [(mean ± standard deviation (SD) increase NT-proBNP, 38.1 (±37.7) vs. 5.9 (±12.3) pg/mL, *p* < 0.001] [84]

A recent systematic review identified three studies in AECOPD patients that have high negative predictive values (0.80–0.98) to exclude left ventricular dysfunction [66] with different cut-off values for pro-BNP (500 pg/L) or NT-pro-BNP (935 or 1000 pg/L).

Another author found in COPD subjects who had chest X-rays at the time of hospitalization for AECOPD that NT-proBNP was a good indicator of radiological parameters related to cardiac dysfunction and/or volume overload such as cardiac size, pulmonary edema, and pleural effusion with AUCs of 0.72, 0.63, and 0.64, respectively [67].

COPD is a frequent comorbid condition in patients with community-acquired pneumonia (CAP) with reported rates of between 15% and 42% [85,86,87]. On the other hand, the incidence of pneumonia in COPD patients is almost twice that of the general population [88]. Although some guidelines have included pneumonia as one of the causes of exacerbation of COPD, it is currently considered an infectious comorbidity, and therefore must be differentiated from AECOPD. The differential diagnosis between CAP and AECOPD is traditionally based on the radiological condensation. However, the usefulness of biological markers, particularly CRP and PCT, has been investigated in order to differentiate between AECOPD and CAP, and to identify bacterial infections that could benefit from antibiotic treatment. Significant differences in the CRP and PCT levels have been reported between patients with CAP and COPD compared with patients with AECOPD. Huerta et al. showed that AECOPD and CAP differed in clinical and inflammatory expression, with a greater increase in biological markers such as CRP and PCT in CAP, as well as a higher incidence of fever, chills, pleuritic pain and crepitus [89]. The study also showed increased serum levels of tumor necrosis factor-α, and interleukin (IL)-6, IL-1 and IL-8 in patients with CAP and COPD compared with patients with AECOPD [89]. Another study reported that CRP, PCT, and neopterin (NPT) levels were significantly increased in patients with CAP compared with the patients with AECOPD, whereas the CRP/NPT ratio was lower [90]. The CRP/NPT ratio was considered to discriminate between AECOPD, CAP and CAP with COPD, with a cutoff ratio of 0.346 (sensitivity, 65% and specificity, 79%) [90].

## 6. Toward a More Precise Definition of COPD Exacerbation

In a recent study that includes subjects from two large cohorts [SPIROMICS (Subpopulations and Intermediate Outcomes Measures in COPD Study) and COPDGene (Genetic Epidemiology of COPD)], Keene et al. [91], analyzed a panel of plasma biomarkers previously reported, and linked with the presence of AECOPD in the past 12 months. Surprisingly, the authors found that despite the reported correlation of several biomarkers level with the exacerbation risk, there was a poor replication between cohorts. Moreover, biomarkers added little to the predictive power in comparison with clinical parameters to determinate the risk of AECOPD. One possible explanation for these findings could be that the heterogeneity of the etiology of AECOPD was not taken into account in the analysis. These results highlight the need for more accurate methods to identify the different triggers that determinate the activation of diverse pathological pathways in an AECOPD.

More recently, Noell et al. [92], in a proof-of-concept study, evaluated 86 hospitalized patients with AECOPD in a multicenter trial, and fully characterized the patients up to three months. They assessed clinical, biological, microbiological, functional and radiological variables, and built a multi-level correlation with the different parameters. The findings showed that a panel of biomarkers that includes dyspnea (≥5 on an analogue visual score from 0 to 10), CRP level (≥3 mg/L) and ≥70% circulating neutrophils had a high predictive value for AECOPD diagnosis (AUC 0.97). These parameters are common in several conditions, so, it has to be interpreted within a clinical context considering the differential diagnosis.

Furthermore, it is well established that early recognition of symptoms and the application of a prompt therapy in AECOPD patients leads to a faster recovery, reduces risk of hospitalization, and is associated with a better quality of life [29]. For the above-mentioned, the need to develop a precise diagnostic algorithm for AECOPD that includes symptoms and biomarkers as a network approach is a step forward that should be taken.

As the different studies described above support, the inclusion of the symptoms within this algorism of exacerbation definition is unquestionable. However, not only the presence of symptoms should be incorporated within the algorism, but also a precise level of “symptom’s worsening” in particular of dyspnea based on the results of the recent studies. Moreover, the same studies endorse the use of objective measure biomarkers that define relevant changes in symptoms related with AECOPD, such as dyspnea. In the same line, Celli BR [93] estimated the need of a more precise definition of AECOPD, made a proposal with a more holistic approach that included several clinical, biological as well as differential diagnostic parameters. 

The symptom-based and healthcare-based definitions are pragmatic approaches that are widely used, however, they often oversimplify the pathogenic pathways implicated in an AECOPD, so it is also necessary to include markers in the algorithm that help in the etiological diagnosis of the exacerbation or in the identification of different exacerbation phenotypes to improve precise therapeutic approaches. It is also important to incorporate markers that help differentiate AECOPD from other respiratory processes with similar clinic, such as pneumonia.

In the light of these evidences, we propose a simple definition for AECOPD that, in addition to the respiratory symptoms includes APRs biomarkers (CRP and neutrophils), etiological biomarkers (PCT), and biomarkers for differential diagnosis (pro BNP or NT-pro-BNP, and X Rays). The suggested biomarkers to be included in the proposed definition of AECOPD are: CRP ≥3 mg/L and ≥70% circulating neutrophils in the diagnosis of AECOPD; PCT as an indicator of bacterial infection and therefore for the use antibiotics; and pro-BNP as a marker for cardiac dysfunction (differential diagnosis). Table 3 shows the proposal towards a more precise definition of AECOPD following the conclusion of the data analyzed in the present review. It is important to highlight that it is only a proposal and this simple definition has not been validated in any previous study, therefore its use in clinical trials or in daily practice is not recommended until its usefulness has been validated.

## 7. Conclusions

In the present review we have analyzed the weaknesses and strengths of the different used AECOPD definitions, as well as the findings of the studies that have assessed the relationship of different biomarkers with the diagnosis, etiology and differential diagnosis of AECOPD. We have also analyzed the impact of the AECOPD definition on the assessment of the treatment effect from clinical trials. These results emphasize the need for concerted attempts to reach a consensus on a more precise and objective definition of AECOPD. On the basis of the available evidence, we also agree that it is time to stop defining an AECOPD only by its symptoms or the healthcare resources used. Therefore, we have proposed a simple definition of AECOPD, which must be validated in future clinical trials to define its accuracy and potential usefulness in daily practice.

## Figures and Tables

**Table 1 medsci-06-00050-t001:** Examples of chronic obstructive pulmonary disease (COPD) exacerbation definitions based on the presence of symptoms (symptom-based definition), the types of healthcare resources used (healthcare-based definition), or the combinations of both.

Study	Definition	Basis of Definition	
		Symptom	Healthcare
Anthonisen, N.R. et al. [12].	Type 1: Occurrence of increased dyspnea, sputum volume and sputum purulence. Type 2: Two of these symptoms were present. Type 3: One of the three symptoms was present plus at least one of the following: upper respiratory infection within the past five days, fever without other cause, increased wheezing or cough, increase in respiratory rate or heart rate by 20%.	X	
Rodriguez-Roisin, R. [13]	Sustained worsening of the patient’s condition from the stable state and beyond normal day-to-day variations, which is acute in onset and necessitates change in regular medication.	X	X
Vogelmeier, C.F. et al. (GOLD 2017) [5]	An acute worsening of respiratory symptoms that results in additional therapy.	X	X
Miravitlles, M. et al. (GesEPOC guideline) [14]	A clinical episode occurring during the course of COPD, characterized by a sudden or gradual worsening of symptoms that is beyond expected daily variability and cannot be attributed to other disorders.	X	
Wedzicha, J.A. et al. (ERS/ATS guideline) [15]	Episodes of increasing respiratory symptoms, particularly dyspnea, cough and sputum production, and increased sputum purulence.	X	
Burge, P.S. et al. (ISOLDE study) [16]	Worsening of respiratory symptoms that require oral corticosteroids or antibiotics or both.		X
Mahler, D.A. et al. [17]	Defined by treatment (mild: increased use bronchodilator; moderate: use of antibiotics and/or corticosteroids; severe: hospitalization).		X
Szafranski, W. et al. [18]	Severe: requirement for oral steroids and/or antibiotics and/or hospitalization due to respiratory symptoms. Mild: a day with ≥4 inhalations of reliever medication above the mean run-in use.		X
Calverley, P. et al. [19]	Worsening of COPD symptoms that required treatment with antibiotics, oral corticosteroids, or both.		X
Vogelmeier, C. et al. (POET study) [20]	An increase in or new onset of more than one symptom (cough, sputum, wheezing, dyspnea or chest tightness), with at least one symptom lasting three days or more and leading the patient’s attending physician to initiate treatment with systemic glucocorticoids, antibiotics or both (moderate exacerbation) or to hospitalize the patient (severe exacerbation).	X	X
Magnussen, H. et al. (WISDOM study) [21]	Moderate: an increase in lower respiratory tract symptoms related to COPD or the new onset of two or more such symptoms, with at least one symptom lasting three or more days and for which the treating physician prescribed antibiotics, systemic glucocorticoids or both. Severe: an exacerbation requiring admission to hospital.	X	X
Wedzicha, J.A. et al. (FLAME study) [22]	Mild (involving worsening of symptoms for >2 consecutive days but not leading to treatment with systemic glucocorticoids or antibiotics), moderate (leading to treatment with systemic glucocorticoids, antibiotics or both) or severe (leading to hospital admission or a visit to the emergency department that lasted >24 hours in addition to treatment with systemic glucocorticoids, antibiotics or both)	X	X

GOLD: Global Initiative on Chronic Obstructive Lung Disease; GesEPOC: Spanish guidelines for diagnostic and treatment of COPD; ERS/ATS: European Respiratory Society/American Thoracic Society; ISOLDE: inhaled steroids in obstructive lung disease in Europe; POET: prevention of exacerbations with tiotropium; WISDOM: withdrawal of inhaled steroids during optimized bronchodilator management; FLAME: indacaterol/glycopyrronium versus salmeterol/fluticasone for COPD exacerbations.

**Table 2 medsci-06-00050-t002:** Biomarkers in acute exacerbations in COPD according to clinical use.

Biomarker	Definition and Risk Assessment	Etiology of AECOPD	Differential Diagnosis
CRP *	Elevated in AECOPD [36,37,38]. Related with the AECOPD severity [38]. Combined with plasma pro-BNP level identify patients that need hospitalization [39].		
WBC *	Levels of 9 × 10^9^/L associated with increased risk of exacerbations [6,40].		
Fibrinogen *	Elevated in AECOPD [41,42]. Associated with increased risk of AECOPD [43,44] and mortality [45].		
IL-6	Elevated in AECOPD [42,46].	Increased in ACOPD by viral etiology [47].	
Serum uric acid	Increased risk of AECOPD, hospitalization and use of non-mechanical ventilation [48].		
SP- D	Increased in AECOPD and inversely related to FEV_1_ [49].		
CCL-18	Elevated in AECOPD and associated with risk of exacerbation that requires hospitalization [50].		
VIP	Increased levels are diagnosis of AECOPD [51].		
Copeptine	Increased in AECOPD. Associated with prolonged hospitalization and treatment failure [52].		
E-selectine	Increased in frequent exacerbators [53].		
MMP9/TIMP-1	Increased during AECOPD and negatively correlated with spirometric variables [54,55,56].		
PTX- 3		Correlated with bacterial isolation in sputum [57].	
IP-10		Increased levels correlated with presence of human rhinovirus load in sputum [58].	
soluble IL-5 receptor α		Increased in AECOPD due to viral infection [59].	
MCP-1		Correlated with the presence of viral infection [46].	
TNF α		Sputum levels correlated with bacterial isolation [60]. Plasma levels are associated with viral infection [46].	
IL-1β		Sputum levels associated with bacterial isolation [61].	
Procalcitonin		Use as a guide for antibiotic treatment in bacterial AECOPD [62].	
Eosinophils		Related with lower bacterial isolation [63]. Used to direct corticosteroid therapy in AECOPD [64,65].	
Pro-BNP/NT pro-BNP			With cardiac diseases [66,67].

CRP: C-reactive protein; WBC: white blood cells; IL-6: interleukin-6; SP-D: surfactant protein D; CCL-18: chemokine (C-C motif) ligand 18; VIP: vasoactive intestinal peptide; MMP9/TIMP-1: metalloproteinase 9/tissue inhibitor of metalloproteinase protein-1; PTX-3: pentraxin-3; IP-10: interferon γ-induced protein-10; IL-5: interleukin-5; MCP-1: monocyte chemotactic protein-1; TNF α: tumor necrosis factor α; IL-1β: interleukin-1β; proBNP: pro brain natriuretic peptide; NT proBNP: N-terminal-pro brain natriuretic peptide. * Plasma levels of CRP combined with WBC plus fibrinogen showed increased risk of exacerbations [40].

**Table 3 medsci-06-00050-t003:** A new proposal towards a more precise definition of AECOPD.

Parameter	Value
Dyspnea	≥5 (VAS) *
Plasma CRP (mg/dL)	≥3 mg/dL
Blood neutrophils (%)	≥70%
Procalcitonin (μg/L) **	>0.25
NT-ProBNP (pg/mL) ^†^	>300
X-Rays	No pneumonia

* VAS: visual analogue scale from 0–10. ** >0.25 μg/L suggest a bacterial etiology and encourage the use of antibiotic therapy. ^†^ >300 pg/mL suggest cardiac dysfunction.

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
