# Peer review of "Is It Time to Change the Definition of Acute Exacerbation of Chronic Obstructive Pulmonary Disease? What Do We Need to Add?"

_medsci, 2018, doi:10.3390/medsci6020050_

Round 1

Reviewer 1 Report

I think it's a very correct article on a very current and hot topic. we need to define correctly the exacerbations of the epoc in order to treat them in a correct and defined way. The use of biomarkers to differentiate one from the other is a challenge that we must solve.

Author Response

We thanks the reviewer for your comments. 

Reviewer 2 Report

Major comments

1)      This is a nicely written review of AECOPD definitions. There is however a very large omission in the manuscript that describes symptom-based definitions of AECOPD. Patient-reported outcomes such as the EXACT are not mentioned at all. PROs provide validated symptom-based definitions of exacerbations. In fact PRO could eliminate many of the issues with other symptom-based and HCRU definitions.  This omission must be corrected, not only with descriptions and examples of their use in clinical trials but with reference to their prominence in FDA and EMA guidance for RCT.

2)      More detail should be added regarding the strengths and weaknesses of HCRU which is given only a brief description at present (lines 152-4).

3)      “Chen et al.37, found that plasma CRP levels combined with pro- Brain 261 Natriuretic Peptide (pro-BNP) level could identify patients that need hospitalization 262 during an AECOPD.”

This is incorrect. The study compared patients hospitalized for AECOPD vs control but the level of biomarker was not examined against outpatient exacerbations so it cannot be reliably inferred that this combination of biomarkers accurately reflects the need for hospitalization, an outcome that varies massively from one healthcare setting to another. At most this combination may be used to identify patients having an exacerbation.

4)      It is unsurprising that thus far a single/combination of biomarkers has been able to provide any real meaningful data in the field of exacerbation assessment. Exacerbations are varied in aetiology, presentation and consequence, in addition all these aspects vary between patients of differing severity. In truth we should be trying to phenotype exacerbations as well as patients and adjust preventative and acute treatment accordingly. This should be explored in much more detail in the manuscript.

5)      The proposed new definition of AECOPD is completely unproven and appears to add little to current biomarker definitions especially the Noell et al study. It should be removed as at present no justification has been provided for why adding a few more elements will add value. If the authors wish to include this they should at least assess this is a retrospective database first and provide data to assess sensitivity and specificity for diagnosis prior to requesting its examination in future clinical trials.

6) The DECAF score for some reason is not mentioned or discussed even though this combination of biomarkers and clinical information has been proven to provide meaningful prognostic information and recently been shown to add value to clinical practice. This should be corrected.

7) It would be useful to discuss the controversy regarding AECOPD in the presence of pneumonic features on CXR and some of the biomarker work in this area.

Minor comments

Table 2-CRP definition and risk assessment-typo Related with “de” AECOPD severity

Author Response

We are grateful for your thorough review. We carefully considered all of your comments and have responded to each of them below.

1)      This is a nicely written review of AECOPD definitions. There is however a very large omission in the manuscript that describes symptom-based definitions of AECOPD. Patient-reported outcomes such as the EXACT are not mentioned at all. PROs provide validated symptom-based definitions of exacerbations. In fact PRO could eliminate many of the issues with other symptom-based and HCRU definitions.  This omission must be corrected, not only with descriptions and examples of their use in clinical trials but with reference to their prominence in FDA and EMA guidance for RCT.

Response: We appreciate the review observation and we fully agree with the need to incorporate this information in the new version. Following the reviewer recommendations, we have added a comment in the revised version regarding the use of PROs in the validation of symptom-based and HCRU definitions. 

2)      More detail should be added regarding the strengths and weaknesses of HCRU which is given only a brief description at present (lines 152-4).

Response: Following the reviewer recommendations, we have added a comment in the revised version regarding the strengths and weaknesses of HCRU and the incorporation of information collected directly from patients with standardized instruments such as PROs to improve the accuracy of AECOPD definition based on events.

3)      “Chen et al.37, found that plasma CRP levels combined with pro- Brain 261 Natriuretic Peptide (pro-BNP) level could identify patients that need hospitalization 262 during an AECOPD.”

This is incorrect. The study compared patients hospitalized for AECOPD vs control but the level of biomarker was not examined against outpatient exacerbations so it cannot be reliably inferred that this combination of biomarkers accurately reflects the need for hospitalization, an outcome that varies massively from one healthcare setting to another. At most this combination may be used to identify patients having an exacerbation.

Response: We understand the reviewer concern regarding the interpretation of the Cheng et at., findings, therefore we modified the text in order to be more precise highlighting that the combination of two biomarkers could separate patients who were experiencing AECOPD that required hospitalization from stable patients.

 4) It is unsurprising that thus far a single/combination of biomarkers has been able to provide any real meaningful data in the field of exacerbation assessment. Exacerbations are varied in aetiology, presentation and consequence, in addition all these aspects vary between patients of differing severity. In truth we should be trying to phenotype exacerbations as well as patients and adjust preventative and acute treatment accordingly. This should be explored in much more detail in the manuscript.

Response: We totally agree with the reviewer opinion and in the same line we have extensible describe the role of biomarkers by etiology (pauci-inflammatory, bacterial, viral, eosinophilic). Therefore we consider that the topic is well covered with the provided information in the sections of “Biomarkers associated with the etiology of exacerbation”, and “Toward a more precise definition of COPD Exacerbation”.

5) The proposed new definition of AECOPD is completely unproven and appears to add little to current biomarker definitions especially the Noell et al study. It should be removed as at present no justification has been provided for why adding a few more elements will add value. If the authors wish to include this they should at least assess this is a retrospective database first and provide data to assess sensitivity and specificity for diagnosis prior to requesting its examination in future clinical trials.

Response: We understand the reviewer's concern about the unproven new definition of exacerbation. However, we believe that according to the data shown in the review, there is a need to move forward with a simple new definition of exacerbation. Raising a new proposal without prior validation is nothing new. In fact, the document GOLD 2011 and 2017 propose a new classification of COPD without previous validation, with the purpose that these were submitted later to analysis in different clinical settings. For this reason, we have decided to maintain our approach, making very clear that this proposal has not been validated and therefore its use in clinical trials or in daily practice is not recommended until its usefulness has been validated.

6) The DECAF score for some reason is not mentioned or discussed even though this combination of biomarkers and clinical information has been proven to provide meaningful prognostic information and recently been shown to add value to clinical practice. This should be corrected.

Response: We totally agree with the reviewer with the value of the DECAF score to predict the hospital mortality in AECOPD, however this score is beyond the objective of the present review, due to this review is fundamentally focus in the analysis of the definitions of acute exacerbations and the biomarkers that could help identify these events and its differential diagnostic. For these reasons we have decided not to included the DECAF in the revised version of the manuscript. These data should be included in other documents aims to asses the prognosis or mortality prediction associated to acute exacerbations of COPD.

7) It would be useful to discuss the controversy regarding AECOPD in the presence of pneumonic features on CXR and some of the biomarker work in this area.

Response: Following the reviewer recommendations, we have added a comment in the revised version regarding the use of biomarker in the differential diagnosis between pneumonia and AECOPD.

Minor comments

Table 2-CRP definition and risk assessment-typo Related with “de” AECOPD severity

Response: corrected

Round 2

Reviewer 2 Report

1)     It is pleasing that the authors have incorporated a section on PROs but the final paragraph requires significant modification/deletion.

Although PROs seem to be useful tools to improve the accuracy of the AECOPD 155 definitions, they have some limitations that should be highlighted. In general, patients 156 with COPD are older adults and frequently they have difficulties with the management of 157 complex technology. In addition, the viability and availability of these instruments are 158 limited for patients with low resources in particular those from underdeveloped countries. 159 Therefore, it is unlikely that they can be widely used in clinical practice worldwide and 160 used as simple AECOPD definition.

PROs, and the accompanying technology to deliver them, are not complex, the consist of simple responses to a limited number of questions, 14 in the case of the EXACT that is administered via mobile phones or tablets. The authors proposal that elderly patients cannot use such technology is false, discriminatory and has not been substantiated with a reference. This prejudicial statement that should be removed. The EXACT was validated in a standard COPD population and the fastest growing users of smart phones and mobile apps are older populations where up to 75% of people use digital technology daily. The suggestion that such technology is not viable in underdeveloped countries is also false due to the mass penetration of this technology in such settings. More people in Africa have access to cell phone service than have piped water. Furthermore the delivery of PROs is not necessarily dependent upon technology since both the EXACT and other PROs such as daily symptom diary cards and the COPD assessment test have been successfully delivered via paper.

The limitations of PROs in clinical practice surround mainly copyright ownership for instruments such as the EXACT that would preclude mass usage due to cost, and fatigue from patients that might reduce completion rates if they were used over long periods of time.

2)     The HCRU section is well-written and a welcome addition.

3)     Chen amendment is fine.

4)     Fine

5)     I disagree with the response and would again recommend removal of the unproven exacerbation definition proposal. The example given by the authors proves the problems that accompany unvalidated proposals since the GOLD definition has been revised multiple times since its publication, has been shown to be inferior to prior definition in predicting core outcomes such as mortality and it continuously a source of controversy and disagreement in the field.

6)     Fine

7)     The new CAP section is nicely written.

Author Response

1)     It is pleasing that the authors have incorporated a section on PROs but the final paragraph requires significant modification/deletion.

Although PROs seem to be useful tools to improve the accuracy of the AECOPD 155 definitions, they have some limitations that should be highlighted. In general, patients 156 with COPD are older adults and frequently they have difficulties with the management of 157 complex technology. In addition, the viability and availability of these instruments are 158 limited for patients with low resources in particular those from underdeveloped countries. 159 Therefore, it is unlikely that they can be widely used in clinical practice worldwide and 160 used as simple AECOPD definition.

PROs, and the accompanying technology to deliver them, are not complex, the consist of simple responses to a limited number of questions, 14 in the case of the EXACT that is administered via mobile phones or tablets. The authors proposal that elderly patients cannot use such technology is false, discriminatory and has not been substantiated with a reference. This prejudicial statement that should be removed. The EXACT was validated in a standard COPD population and the fastest growing users of smart phones and mobile apps are older populations where up to 75% of people use digital technology daily. The suggestion that such technology is not viable in underdeveloped countries is also false due to the mass penetration of this technology in such settings. More people in Africa have access to cell phone service than have piped water. Furthermore, the delivery of PROs is not necessarily dependent upon technology since both the EXACT and other PROs such as daily symptom diary cards and the COPD assessment test have been successfully delivered via paper.

The limitations of PROs in clinical practice surround mainly copyright ownership for instruments such as the EXACT that would preclude mass usage due to cost, and fatigue from patients that might reduce completion rates if they were used over long periods of time.

Response: We accept the review comment and have modified the text following the limitations suggested by the reviewer

2)     The HCRU section is well-written and a welcome addition.

Response: Thanks

3)     Chen amendment is fine.

Response: Thanks

4)     Fine

Response: Thanks

5)     I disagree with the response and would again recommend removal of the unproven exacerbation definition proposal. The example given by the authors proves the problems that accompany unvalidated proposals since the GOLD definition has been revised multiple times since its publication, has been shown to be inferior to prior definition in predicting core outcomes such as mortality and it continuously a source of controversy and disagreement in the field.

Response: We understand the reviewer's concern about making a definition proposal to be validated in future studies. We insist that this is a way commonly used in clinical research and another recent example of this practice is the proposal for early COPD (At the root: defining and halting progression of early chronic obstructive pulmonary disease. Martinez F, et al AJRCCM Articles in Press. Published on 06-February-2018 as 10.1164/rccm.201710-2028PP).  Therefore, we have decided to keep the proposal in the document and leave the final decision on the point to the editor.

6)     Fine

Response: Thanks

7)     The new CAP section is nicely written.

Response: Thanks
